# Microbial and Fungal Phytases Can Affect Growth Performance, Nutrient Digestibility and Blood Profile of Broilers Fed Different Levels of Non-Phytic Phosphorous

**DOI:** 10.3390/ani10040580

**Published:** 2020-03-30

**Authors:** Youssef A. Attia, Fulvia Bovera, Francesco Iannaccone, Mohammed A. Al-Harthi, Abdulaziz A. Alaqil, Hassan S. Zeweil, Ali E. Mansour

**Affiliations:** 1Arid Land Agriculture Department, Faculty of Meteorology, Environment and Arid Land Agriculture, King Abdulaziz University, P.O. Box 80208, Jeddah 21589, Saudi Arabia; malharthi@kau.edu.sa; 2Department of Veterinary Medicine and Animal Production, University of Napoli Federico II, via F. Delpino 1, 80137 Napoli, Italy; 3Department of Animal and Fish Production, King Faisal University, Al-Hufof, Al-Hassa 31982, Saudi Arabia; aalaqil@yahoo.com; 4Department of Animal and Fish Production, Faculty of Agriculture, Saba Basha, Alexandria University, Alexandria 21527, Egypt; hszeweil@yahoo.com; 5Ministry of agriculture, Animal Production section, Behiri Governorate 22951, Egypt; alierfatmansour@gmail.com

**Keywords:** phytase, Sasso broiler, non-phytic phosphorous, productive performance, blood profiles

## Abstract

**Simple Summary:**

To reduce the environmental pollution is a must to preserve the health of the world. The environmental impact of poultry farming is receiving an increasing attention due to several emissions among these is phosphorus. This element is in general present in the commercial diets of broilers or laying hens in an amount exceeding the real needing of the animals, and, therefore, a great amount of phosphorus ends in the excreta. Thus, optimizing the amount of phosphorous in the diets of poultry could partially alleviate the environmental impact of these farms.

**Abstract:**

A total of 420 day old chicks were divided into seven groups (5 replicates of 12 chicks/group) fed isoproteic and isoenergetic diets. The control group was fed diets containing 0.50%, 0.45% and 0.40% of non-phytic phosphorous (nPP) in starter (1–35), grower (37–56) and finisher (57–64 d) periods, respectively. The three intermediate nPP (IntnPP) groups were fed diets with 0.40%, 0.35% and 0.30% nPP according to the growth period and were submitted to three dietary treatments: unsupplemented; supplemented with 500 FTU/kg diet of an *Aspergillus niger* phytase (IntnPP_fp) and supplemented with 500 FTU/kg diet of an *Escherichia coli* phytase (IntnPP_bp). The three low nPP groups fed diets contained 0.30%, 0.25% and 0.20% nPP and were submitted to the same dietary treatments than IntnPP to obtain LnPP, LnPP_fp and LnPP_bp groups. IntnPP and LnPP groups had lower body weight gain and feed, crude protein (CP) and metabolizable energy (ME) intake (*p* < 0.05) than the control. Feed conversion ratio of IntnPP was more favorable (*p* < 0.01) than the LnPP group. CP and ME conversion ratios worsened (*p* < 0.01) in IntnPP and LnPP groups in comparison to the control. The nPP conversion ratio improved (*p* < 0.01) from the control to the LnPP group. Fungal phytase reduced (*p* < 0.05) feed, CP, ME and nPP intake than the bacterial one. IntnPP and LnPP diets had a lower digestibility of CP (*p* < 0.01) and CF (*p* = 0.01) than the control. IntnPP and LnPP groups showed a higher (*p* < 0.05) economic efficiency than the control. Blood total protein was the lowest (*p* < 0.05) in the LnPP group, the control group showed the lowest (*p* < 0.05) level of albumin and IntnPP group had the lowest (*p* < 0.01) globulin level. The use of bacterial phytase increased (*p* < 0.01) total protein and globulin and decreased (*p* < 0.05) the plasma cholesterol in comparison to fungal phytase. Decreasing nPP levels in colored slow-growing broilers diet negatively affects growth performance and the use of phytase can partly alleviate these negative effects, but the efficiency of different enzyme sources (bacterial or fungal) was tied to the dietary nPP levels.

## 1. Introduction

Phosphorus (P) is an essential nutrient for plants and animals and is critically important in the production of poultry. Phosphorus is well represented in the vegetables ingredients commonly used in poultry diet preparation (as soybean or corn), however 50%–80% of total phosphate in plant seeds is stored as phytate P [1]. Phosphorous phytate is poorly available to intestine absorption of monogastric animals and can also reduce the digestibility of other nutrients as well as the performance of animals owing to its antinutritional effect [2,3]. It is possible to add a further amount of non-phytic P to the diet, but this allows an increase of the amount of P released, especially from the waste of intensive poultry farms. It is well known that excess of P in the water led to its eutrophication and the subsequent enrichment of surface water by plant nutrients is a form of pollution [4]. This problem has led to numerous studies to reduce the P amount in the waste from the poultry industry. Several studies indicated great differences in the non-phytate phosphorus (nPP, inorganic) requirement of broilers compared to the data presented in the 1994 by the National Research Council (NRC). Waldroup et al. [5] showed that the nPP requirement for the starter phase of broilers ranges from 0.37% to 0.39%. Angel et al. [6,7] fixed the nPP requirement of the broiler between 0.32% and 0.28% in the growing period (18–32 d), 0.24% and 0.19% in the finisher period (32–42 d) and 0.16% and 0.11% in the withdrawal period (42–49 d). Dhandu and Angel [8] reported a requirement of 0.20% nPP for the finisher period and 0.16% nPP in the withdrawal period of broiler. All these nPP requirements are substantially lower than that reported by the NRC [9]. These data suggested that it is possible to reduce the P concentration in the diet and this can reduce the amount of P in the manure without negative effects on poultry performance. The addition of phytase to the diet can help to release P from phytate and thus reduce the amount of non-phytic P supplemented to the diets. Phytase is the most common enzyme used in the feed for monogastric animals: it can reduce the antinutritional effect of phytate and improve the digestibility of phosphorous, calcium, amino acids and energy, as well as reduce the negative impact of inorganic P excretion to the environment [10]. Phytase are phosphatases hydrolyzing one or more phosphate group. Depending on the position of the phosphate group on the myo-inositol ring, which they firstly hydrolyze, they belong to one of two subclasses, 3-phytase and 6-phytase [11]. Ghazalah et al. [12] and Selim [13] indicated that phytase supplementation to the diet of fast-growing chickens can replace around 0.1% of dietary P; however, the data are inconsistent due to type of diet, dietary composition, type and age of chicks, type and dose of phytase. So, there are many areas of research that need to be addressed such as the effect of type of phytase and the efficacy of bacterial (phytase-6) vs. fungal phytase (phytase-3) on phosphorus utilization, due to the exist contradictory results [6,14,15,16,17,18].

The aim of our research was to study the effect of microbial or fungal phytase on growth performance, nutrient digestibility, carcass and meat quality and blood profile of colored slow-growing Sasso broilers fed diets with different supplementation of non-phytic phosphorous.

## 2. Materials and Methods 

### 2.1. Chickens, Experimental Design and Husbandry

The department committee of Animal and Poultry Production accepts all the procedures done in the current study. These procedures suggest minimal stress to the animal to ensure rights and welfare by eliminating harm or suffering to animals according to the official decrees of the Ministry of Agriculture in Egypt regarding animal welfare (Decree No. 27 (1967) that enforces the humane treatment of animals generally).

A total of 420 unsexed day-old colored broilers (Sasso strain) were homogeneously divided into 7 groups (5 replicates of 12 chicks each). Chicks were housed in 35 floor pens (1.0 m × 1.0 m/pen) furnished with rice hulls as a litter. Along the experimental period, chicks were fed three different diets, according to the animal age: starter (1–35 d of age); grower (36–56) and finisher diet (57–64). The experimental period was listed from 1 to 64 d of age. The control group fed diets supplemented with 0.50%, 0.45% and 0.40% of non-phytic phosphorus (nPP), respectively for each period. The three groups fed intermediate nPP (IntnPP) diets supplemented with 0.40%, 0.35% and 0.30% nPP in the starter, grower and finisher periods, respectively were submitted to three dietary treatments: unsupplemented, supplemented with 500 FTU of a fungal phytase (IntnPP_fp, BASF Germany, n-3 Aspergillus niger phytase)/kg diet and supplemented with 500 FTU of a bacterial phytase (IntPP_bp, Phyzyme^®^ XP, n-6 *Escherichia coli* phytase, Danisco Animal Nutrition)/kg diet. The three groups fed low nPP (LnPP) diets were supplemented with 0.30%, 0.25% and 0.20% nPP in the starter, grower and finisher periods, respectively and were submitted to the same dietary treatments than IntnPP groups to obtain LnPP, Lnpp_fp and LnPP_bp fed groups.

The experimental diets (Table 1) were formulated according to NRC [9]. Washed building sand was used at small amount (0.14%–0.50%) to keep nutrients profiles of different diets similar as far as possible as, otherwise the change in feedstuffs contents may affect nutrients profiles. 

Feed samples of the starter, grower and finisher diets were chemically analyzed for dry matter, crude protein, ether extract, crude fiber and ash according to the Association of Official Agricultural Chemists (AOAC) [19] official methods. The nitrogen free extract was determined by subtracting the sum of all fractions mentioned above from the dry matter (DM).

Mash form diets and clean water were offered ad libitum. Birds were illuminated with 24 h light cycle during the first 3 days of age and then with 23:1 light-dark cycle, according to Attia et al. [20]. The vaccinations and medical care were conducted according to veterinary indications. Chicks were farmed under similar environmental, managerial, and hygienic condition and breed according to Hendrix Genetic indication for the Sasso strain (available at: https://www.hendrix-genetics.com/en/news/new-brand-promise-sasso/).

### 2.2. Data Collection 

Birds were weighed (g) in the morning before offering feeds at the beginning (day 1) and at the end of the trial (day 64) to calculate body weight gain for each replicate. Each group was provided daily with enough pre-weighed feed of its corresponding diet. The remainder and scattered feed as well as the consumed feed were weekly calculated for each replicate to calculate feed intake in the periods 1–64 days of age. The consumption of CP, ME and nPP was calculated by multiplying the CP, ME and nPP contents of the experimental diets by its corresponding feed consumption during the entire experimental period.

Conversion indexes of feed (feed conversion ratio (FCR), g feed/g gain), crude protein (crude protein conversion ratio, CPCR, g protein/g gain) and ME (metabolizable energy conversion ratio, MECR, MJ/g gain) of the diets were calculated during the experimental period. Utilization of P as the nPP conversion ratio was calculated as g of nPP consumption required to produce 1000 g body weight gain. Mortality rate was calculated as the percentage of birds dead in each treatment during the entire experimental period.

At 64 days of age, 6 broilers per treatment as three replicates of two males each were used to measure the nutrient digestibility of the diets according to the total collection method described by Attia et al. [20]. The broilers were fasted for 24 h, thus fed on their experimental diets for 120 h; feed intake and excreta were measured during 72 h. The excreta, collected for each replicate, sprayed with 4% boric acid to capture the ammonia in the form of ammonium borate. The excreta were cleaned from feathers and feed, weighed and dried in a forced air oven at 70 °C until constant weight. Samples were finely ground and placed in screw-top glass jars until analyses.

Fecal nitrogen was separated from urine in the excreta samples according to Jakobsen et al. [21]. Total nitrogen, fecal nitrogen, fat, crude fiber and ash contents were determined in the excrement and feeds according to AOAC [19] procedures (dry matter, method number 934.01; crude protein, method number 954.01; ether extract, method number 920.39; crude fiber, method number 954.18 and ash, method number 942.05) and expressed on a dry matter basis. Phosphorus concentrations were determined utilizing flame spectrophotometric techniques using the method of Haraguchi and Fuwa [22].

The apparent digestibility of the nutrients (dry matter, crude protein, fat and fiber) and the apparent retention of ash were calculated by dividing the daily amount retained (g/d) by the amount intake (g/d). The daily amount retained is equal to the amount intake (% nutrient in feed × g of feed consumed) minus that lost in the excreta (% nutrient in excreta × g of excreta voided). Metabolic nitrogen was equal to excreta nitrogen − fecal nitrogen.

At 64 d of age, six birds (3 males and 3 females) were randomly collected from each treatment, weighed after fasting overnight, slaughtered, their feathers were plucked and the inedible parts (head, legs and inedible viscera) were taken aside. Thus, the remaining carcass (dressed carcasses) was weighed and expressed as a percentage of the body weight. The internal organs (liver, pancreas and spleen) were weighed and expressed as percentage of live body weight. Abdominal fat (located in the abdominal cavity, surrounding the intestines and around the heart) was separated, weighed and expressed as a percentage of the live body weight.

A sample of 50% breast muscle + 50% thigh muscle was weighed and kept in an electric drying oven at 70 °C until constant weight. The dried meat was ground to pass through a sieve (1 mm²) and then carefully mixed. The air-dried samples were kept into tight glass container for the subsequent chemical analysis [19]. Protein content of meat was calculated as 100 – moisture − ash − fat. The ability of meat to hold water (WHC), meat tenderness, pH, color intensity and drip loss were measured according to Bovera et al. [23].

The right tibia was removed from the carcass of 6 birds per group, cleaned from tissues, set in hexane for 48 h to remove fat and dry in an oven until constant weight. Their length (mm), width (mm), weight (g) as well as ash, Ca and P contents were determined. Length and width of tibia were determined using a Vernier caliber. The width of tibia was measured at three points in the middle and at the end of both sides. Ca and P contents were determined after ashing at 600 °C, according to the method of AOAC [24].

Blood samples were collected at 64 d of age from 6 broilers per group in heparinized tubes. Plasma was separated by centrifugation at 3000 rpm for 15 minutes and stored at −20 °C until analysis. Concentrations of plasma Ca, inorganic P and alkaline phosphatase were determined according to Attia et al. [25]. All biochemical traits of blood plasma (total protein, albumin, alaninine aminotransferase (ALT), aspartate aminotransferase (AST), total lipids and cholesterol) were determined using commercial diagnosing kits (Diamond Diagnostics Company, Egypt) as reported by Attia et al. [26,27,28]. Globulin concentration was calculated as the difference total protein − albumin.

Economic evaluation for all experimental diets was made. Economic efficiency was calculated as described by Attia et al. [29].

### 2.3. Statistical Analysis

Data were analyzed using the GLM procedure of Statistical Analysis Software (SAS) version 6.11 (SAS Institute Inc.: Cary, NC, USA) [30] by one-way design as the phytases was added to two nPP levels only resulted in an unbalanced experimental deign as the recommended phosphorus levels usually not supplemented with phytases. In addition, the use of factorial analyses in this case can generate main effects that were confounded by the interactions effect. The effect of nPP was compared between the unsupplemented levels and phytases within different nPP levels. Mean difference at *p* ≤ 0.05 was tested using the Student Newman–Keuls test. The replicate was the experimental unit. Data in percentage were transformed to log 10 before running the analyses of variance.

## 3. Results

No deaths were observed in the groups along the trial and all the animals were healthy.

Data on the broilers’ growth performance and feed efficiency in the entire experimental period (1–64 d) are reported in Table 2. The control and IntnPP_bp groups showed higher (*p* < 0.01) body weight gain than the other groups, except for LnPP_fp. The LnPP group showed also lower (*p* < 0.01) body weight gain (BWG) than LnPP_fp group. Feed, crude protein and energy intakes were lower (*p* < 0.01) in LnPP and LnPP_fp than the other groups. nPP intake was higher (*p* < 0.01) in the control, followed by all the IntnPP and then all the LnPP groups. The feed conversion ratio of LnPP group was higher (*p* < 0.01) than the control, IntnPP_bp and LnPP_fp groups. Crude protein and metabolizable energy conversion ratios of LnPP were higher (*p* < 0.01) than the control, IntnPP_bp and _fp and LnPP_fp. IntnPP_bp and LnPP_fp groups showed lower crude protein and metabolizable energy conversion ratios (*p* < 0.01) than the control and LnPP groups.

Table 3 shows the effect of dietary treatments on nutrient digestibility, ash retention and fate of nitrogen in broilers. Both IntnPP and LnPP diets had a lower digestibility of CP (*p* < 0.01) and CF (*p* = 0.01) and a lower (*p* < 0.01) ash retention. Excreta and metabolic nitrogen were higher in IntnPP and LnPP diets in comparison to the control. The effect of the treatments revealed that the use of both phytases improved CP digestibility only when supplemented to IntnPP diets. CF digestibility improved due to the addition of both phytases to IntnPP diets, while fungal phytase supplementation only improved CF digestibility in the LnPP groups. The excreta nitrogen in IntnPP groups decreased (*p* < 0.01) due to use bacterial phytase in comparison to the unsupplemented group while within the LnPP groups the bacterial phytase increased the percentage of excreta nitrogen in comparison to fungal phytase. Fecal nitrogen in the IntnPP and LnPP groups was similar to that of the control and lowered (*p* < 0.01) due to the use of both types of phytase.

Table 4 shows carcass traits and economic indexes of broilers as affected by dietary treatments. LnPP diets had a lower (*p* = 0.01) liver percentage than the other groups. Liver percentage, reduced due to use of fungal phytase in IntnPP groups only.

Tibia characteristics were presented in Table 5. Reducing nPP level in the diets progressively reduced (*p* < 0.05) tibia weight and phosphorus. LnPP diets had a lower (*p* < 0.05) tibia diameter and ash percentage than the other groups, while both IntnPP and LnPP diets reduced (*p* = 0.01) Ca percentage in tibia in comparison to the control. An effect of the treatments was observed for almost all the parameter in the Table except for tibia length. No differences were observed within IntnPP groups for tibia weight, but within LnnPP groups there is a progressive increase of tibia weight from unsupplemented to bacterial and fungal phytase groups with fungal phytase showed stronger effect. Tibia diameter was decreased due to supplementation of fungal phytase in comparison to the other groups in IntnPP diets while both phytases increased tibia diameter in LnPP diets in comparison to the unsupplemented group. Tibia ash was unaffected by dietary treatments in IntnPP groups while both phytases increased (*p* < 0.05) this criterion in comparison to the unsupplemented group when broilers fed LnPP diets. Regarding tibia calcium, no differences were observed among LnPP groups while in IntnPP groups both phytases increased tibia calcium percentage in comparison to the unsupplemented group. Tibia P was unaffected by dietary treatments in IntnPP diets, while in LnPP diets the addition of bacterial phytase increased P percentage in comparison to the unsupplemented group.

No effects of dietary treatments were observed for chemical and physical characteristics of meat (Table 6).

Table 7 shows the effect of dietary treatments on blood criteria. Total protein was the lowest (*p* < 0.05) in LnnP group, the control group showed the lowest (*p* < 0.05) level of albumin and IntnPP group had the lowest (*p* < 0.01) globulin level in blood. Cholesterol was reduced (*p* < 0.05) in IntnPP than the other two groups; the control group showed the highest (*p* < 0.05) level of phosphorus. 

The use of bacterial phytase increased the total protein in comparison to the unsupplemented group when LnPP albumin was detected due to the use of bacterial phytase in IntnPP diets and of fungal phytase in LnPP diets. Bacterial phytase was able to decrease plasma cholesterol in comparison to the other groups when a broiler was fed LnPP diets while no differences were observed among IntnPP groups. Both phytases increased (*p* < 0.05) the phosphorus level in comparison to the unsupplemented group when broilers were fed IntnPP diets while with LnPP diets the addition of fungal phytase increased phosphorus level in comparison to the unsupplemented group.

## 4. Discussion

The decrease of broiler performance due to the reduction of nPP levels in the diets without enzyme supplementation, confirmed the inability of birds to utilize the phytic sources of P, and also confirmed that 0.50, 0.45 and 0.40 nPP levels in the starter, grower and finisher diet (without phytase), respectively were adequate to maintain growth performance.

The worsening of broiler growth performance as weight gain, feed and nutrient intakes are in line with findings by other authors [31,32,33]. According to Karimi et al. [34] the FCR worsened when nPP levels were under 30 mg/kg considering the average value of the three experimental diets. However, the yield of protein and energy was decreased already at the intermediate level of nPP. This result can be ascribed to the combination of low protein and energy intake and a low digestibility of crude protein and fiber found at the intermediate and low nPP levels. Our results disagree with the findings of Karimi et al. [34] who concluded that feeding 0.40% followed by 0.35 % nPP diets during starter and grower periods respectively results in sufficient feed intake of broilers, with such levels actually being in excess of that required for maintaining FCR or mortality rate comparable to the controls. In fact, in our study the reduction of feed intake at the intermediate level of nnP is important and tied to a severe reduction of body weight gain of broilers (−12.27% and −16.50% than the control, respectively for IntnPP and LnPP groups). This discrepancy is probably tied to the age of animals used in our trial. In fact, as a slow-growth genetic type (colored broilers), the slaughter ages of our broilers was 64 days more than double than the slaughter age of broilers used by Karimi et al. [34] in their trial. It is well known that phosphorus requirements are tied to animal age and probably the amounts of nnP at the intermediate and low levels are not sufficient to allow a sufficient storage of the phosphorus in the broilers body, specifically in the bones, with negative effects on animal growth. This is confirmed by a reduction of circulating phosphorus, tibia weight and tibia Ca and P contents observed in our trial with both IntnPP and LnPP diets. In this regard, we must consider that bone density and strength were positively correlated to calcium and phosphorus content in the tibia [35] and that bone characteristics are important in broilers also as indicators of animal health and welfare. The balance between Ca and P in poultry diets is of the utmost importance, particularly in starter diets; Ca: P should be approximately 2:1 [36,37]. In our trial the Ca:P ratio based on calculated and analyzed values of phosphorus in the starter diets was 2:1 vs. 2: 0.94 in the control and 2:0.8 vs. 2:0.76 and 2:0.6 vs. 2:0.54, respectively for IntnPP and LnPP groups. 

The plasma protein showed differences between the different fraction among nPP levels but, very interesting, there is also a difference in albumin to globulin ratio, which is the lowest (0.37) for the control group, while IntnPP and LnPP groups showed similar values (0.82 and 0.70, respectively): high globulin levels and low albumin/globulin ratios indicates better disease resistance and immune response of birds [38].

The results indicate that the effect of type of phytase was correlated with nPP concentrations showing that the differences between the two types of phytase were maximized when the amount of nPP increase in the broiler diet. It is well established that bacterial phytase have a higher efficacy than fungal phytase due to its resistance to low pH an digestive protease [10], so when the amount of nPP in the diet decreases, the bacteria phytase would show a higher efficiency in broiler performance improvement. However, the opportunity to observe improvements in phytase activity is influenced by the level of nPP also in another way: as the nPP level in the diet decreases the efficiency of its utilization increases. In our trial, the phosphorus retention reaches high values and thus the potential to show an improvement in efficiency activity of phytase is greatly reduced. This agrees to other studies in broilers [39,40] and in turkeys [41], which showed that digested P per unit of phytase decreases as the ratio available or non-phytate P/phytase decreases.

This double mechanism of action could be responsible of the results obtained in our trial. As an example, feed, protein and metabolizable energy intakes were not different among unsupplemented, bacterial or fungal phytase groups when nPP level was intermediate, while at the low nPP level bacterial phytase increase all the three criteria when compared to unsupplemented or fungal phytase groups. 

The decrease of CP digestibility due to lower nPP availability is in line with the findings of Xue et al. [42], who observed that, like to pigs, also in poultry there is a linear interrelationship between N and P digestion, quantified by the authors as 10:1. This relationship between protein and P could be linked in various aspects. For example, both muscle and bone developments occur concurrently in growing animals, and both require an adequate amount of AA and P; but also a protein deficiency may affect gut development and therefore the absorption and utilization of P in the small intestine [42]. In general, the researches on this topic focused on the effect of diet protein level modification on P absorption in the developing animals, but as there is a linear correlation, it is also possible that deficiencies of P can affect protein digestion, as shown in our research. A further confirmation is that the use of phytase has a positive impact on CP digestion. As known, phytase has a positive effect on protein digestibility as phytate can bind an amount of free amino acids, such as lysine [43]. In our trial, fungal phytase at the low nPP level was more effective than bacterial phytase on protein availability for metabolic purposes. The increase of feed and nutrient intake in broiler fed bacterial phytase at low nPP levels could be tied to low bioavailability. In this regard, the differences in CP digestibility among low nPP groups were not significant but the group supplemented with fungal phytase had a CP digestibility not different from the groups supplemented with both phytases in broiler fed intermediate nPP levels. This trend to decrease nutrient digestibility as nPP decreases, with beneficial effects due to the use of phytase was observed also for crude fiber and could be ascribed to a higher efficiency of fungal than bacterial phytase. This point needs further study to correlate the phytase activity to different metabolic condition of broilers. On the other hand, our results found a further confirmation in the improvement of nutrient conversion indexes in the fungal phytase low nPP group when compared to the unsupplemented one. However, when low-nPP diets were fed to the broiler, the addition of both phytase had no effects on nPP intake or the conversion ratio.

It is not easy to understand the decrease of the blood cholesterol level in the IntnPP unsupplemented group, but it is very interesting to observe that in broilers that were fed an LnPP diet supplemented with bacterial phytase, the levels of blood cholesterol decreased in comparison to the control and the other LnPP group, suggesting that bacterial phytase had a greater potential than fungal phytase in lowering cholesterol. Several studies [44,45,46] reported that dietary phytate decreased the concentrations of serum total and low-density lipoprotein (LDL) cholesterol in mice or rats. Jariwalla et al. [47] suggested that phytate decreased serum cholesterol by affecting the zinc: copper ratio, as phytate is a strong chelator of divalent cations. Perhaps more likely, the ratio of these cations to each other may affect cholesterol absorption or metabolism. Thus, the addition of phytase can act increasing the cholesterol levels in blood as shown by Liu et al. [48].

The fungal phytase also showed a higher activity than bacterial when low-nPP diets were fed to the broiler as the levels of P in plasma were not different between the two type of phytases at both intermediate and low-nPP diets but, with low-nPP levels fungal phytase was able to increase plasma P levels in comparison to the unsupplemented group. However, at the same nPP levels in the diet, the percentage of P in tibia was not different between phytase levels but bacterial phytase had a higher percentage than the control.

## 5. Conclusions

Decreasing non phytic phosphorous levels in the colored slow-growing broilers diet negatively affects the growth performance and the digestibility of some nutrients (crude protein and fiber). The use of phytase can partly alleviate these negative effects but the efficiency of different enzyme sources (bacterial or fungal) was tied to the nPP levels in the diet. When the amount of non-phytic phosphorous was decreased, the efficiency of P utilization increased and thus in some cases was difficult to appreciate the differences reported in the literature on enzyme efficiency due to the different source and, in other cases, the fungal phytase seemed to show a higher efficiency in comparison to the bacterial phytase. Further investigations need to understand well the interaction between non phytic phosphorus in the broiler diet and efficiency of bacterial or fungal phytase.

## Figures and Tables

**Table 1 animals-10-00580-t001:** Ingredients profiles and nutrient compositions (as the fed basis) of the experimental diets fed during the starter, grower and finisher diets.

Items	Starter	Grower					Finisher		
	0.50 nPP	0.45 nPP	0.35 nPP	0.45 nPP	0.35 nPP	0.25 nPP	0.40 nPP	0.30 nPP	0.20 nPP
Ingredients									
Yellow corn	58.30	58.35	58.35	62.65	62.65	62.70	63.00	63.00	63.55
Soybean meal	32.00	32.00	32.00	27.50	27.60	27.50	28.25	28.25	28.20
Fish meal 65 % CP	3.00	3.00	3.00	3.00	3.00	3.00	-	-	-
Limestone	1.00	1.35	1.73	0.90	1.25	1.60	0.85	1.20	1.55
Dicalcium Phosphate	1.85	1.25	0.65	1.60	1.00	0.35	1.75	1.10	0.50
Vit + Min premix ^1^	0.30	0.30	0.30	0.30	0.30	0.30	0.30	0.30	0.30
NaCl	0.30	0.30	0.30	0.30	0.30	0.30	0.30	0.30	0.30
Methionine	0.15	0.15	0.15	0.15	0.15	0.15	0.15	0.15	0.15
Lysine	0.10	0.10	0.10	0.10	0.10	0.10	0.10	0.10	0.10
Vegetable oils	3.00	3.00	3.00	3.50	3.50	3.50	5.30	5.30	5.30
Washed building sand	0.00	0.20	0.42	0.00	0.15	0.50	0.00	0.30	0.14
Chemical-nutritional characteristics									
Dry matter, % ^2^	89.61	89.83	89.67	89.61	89.83	89.67	89.57	89.72	89.66
CP, % ^2^	21.03	21.04	21.04	19.49	19.42	19.40	17.80	17.83	17.78
ME, MJ/kg ^3^	12.33	12.33	12.33	12.65	12.66	12.65	12.93	12.93	12.97
SAA, % ^3^	0.85	0.85	0.85	0.80	0.80	0.80	0.74	0.74	0.74
Lysine, % ^3^	1.24	1.24	1.24	1.13	1.13	1.13	1.00	1.00	1.00
Calcium, % ^3^	1.00	1.00	1.00	0.90	0.90	0.90	0.80	0.80	0.80
Av. P, %^2^	0.47	0.38	0.26	0.41	0.33	0.23	0.37	0.28	0.17
Ether extract, % ^2^	5.47	5.51	5.45	6.21	6.18	6.22	7.63	7.68	7.59
Crude fiber, % ^2^	3.47	3.47	3.49	3.31	3.36	3.28	3.33	3.36	3.29
Ash, % ^2^	9.24	9.24	9.70	9.30	9.41	9.88	9.18	9.50	9.37
NFE, % ^2^	50.4	50.57	49.99	51.3	51.46	50.89	51.63	51.35	51.63

^1^ Vit+Min mixture provides per kilogram of the diet: vitamin A (retinyl acetate) 24 mg, vitamin E (dl-α-tocopheryl acetate) 20 mg, menadione 2.3 mg, Vitamin D3 (cholecalciferol) 0.05mg, riboflavin 5.5 mg, calcium pantothenate 12 mg, nicotinic acid 50 mg, choline chloride 600 mg, vitamin B12 10 *g, vitamin B6 3 mg, thiamine 3 mg, folic acid 1 mg, d biotin 0.50 mg. Trace mineral (milligrams per kilogram of diet): Mn 80 Zn 60, Fe 35, Cu 8, Se 0.60. ^2^ Analyzed values. ^3^ Calculated values. nPP: non phytic phosphorous; CP: crude protein; ME: metabolizable energy; SAA: sulphate amino-acids; Av. P: available phosphorous; NFE: nitrogen free extract

**Table 2 animals-10-00580-t002:** Growth performance, feed, crude protein, energy intakes and their conversion during days 1-64 of age of Sasso chickens fed diets containing different non-phytate phosphorus levels with or without two types of commercial phytases.

Groups	BWG (g)	FI (g)	CP Intake (g)	ME intake (MJ)	nPP intake (g)	FCR	CPCR	MECR
Control	1769 ^a^	4025 ^a^	794 ^a^	50.78 ^a^	17.17 ^a^	2.36 ^b^	0.449 ^b^	6.86 ^b^
IntnPP	1552 ^bc^	3704 ^a^	732 ^a^	46.76 ^a^	13.16 ^b^	2.39 ^ab^	0.472 ^abc^	7.20 ^abc^
IntnPP_bp	1668 ^a^	3706 ^a^	733 ^a^	46.83 ^a^	13.19 ^b^	2.22 ^b^	0.439 ^c^	6.72 ^c^
IntnPP_fp	1598 ^bc^	3709 ^a^	731 ^a^	46.69 ^a^	13.10 ^b^	2.32 ^ab^	0.457 ^bc^	6.98 ^bc^
LnPP	1478 ^c^	3675 ^b^	726 ^b^	46.37 ^b^	9.37 ^c^	2.49 ^a^	0.491 ^a^	7.50 ^a^
LnPP_bp	1586 ^bc^	3738 ^a^	742 ^a^	47.35 ^a^	9.59 ^c^	2.36 ^ab^	0.468 ^abc^	7.14 ^abc^
LnPP_fp	1646 ^ab^	3689 ^b^	725 ^b^	46.34 ^b^	9.37 ^c^	2.24 ^b^	0.441 ^c^	6.73 ^c^
RMSE	204.2	42.7	8.56	2.15	0.376	0.088	0.0173	0.267
*p* value	0.01	0.01	0.05	0.04	0.0001	0.001	0.001	0.001

^abc^ means within a column under the same treatment with different superscripts are significantly different, NS= not significant, RMSE= root mean square error. BWG = body weight gain; FI= feed intake, CP= crude protein, ME= metabolizable energy, nPP= Non-phytate phosphorus, FCR= feed conversion ratio, CPPR= crude protein conversion ratio, MECR= metabolizable energy conversion ratio; IntnPP: intermediate nPP group; LnPP: low nPP group; bp: bacterial phytase; fp: fungal phytase.

**Table 3 animals-10-00580-t003:** Apparent digestibility, ash retention and excreta and fecal nitrogen of 64-day-old Sasso chickens fed diets containing different non-phytate phosphorus levels with or without two types of commercial phytases.

Groups	Apparent Digestibility, %				Ash Retention, %	Excreta Nitrogen, %	Fecal Nitrogen, %
	DM	CP	CF	EE			
Control	80.6	77.6 ^a^	30.7 ^a^	78.6	31.4 ^a^	4.99 ^b^	2.53 ^a^
IntnPP	81.0	75.3 ^b^	28.8 ^b^	77.4	30.4 ^b^	5.32 ^a^	2.54 ^a^
IntnPP_bp	80.7	77.6 ^a^	31.7 ^a^	77.9	33.5 ^a^	4.05 ^b^	2.33 ^b^
IntnPP_fp	81.2	77.5 ^a^	31.4 ^a^	78.7	33.1 ^a^	5.00 ^ab^	2.30 ^b^
LnPP	80.4	75.3 ^b^	28.6 ^b^	77.7	30.9 ^b^	5.29 ^ab^	2.50 ^a^
LnPP_bp	82.2	75.5 ^b^	29.0 ^ab^	79.6	34.1 ^a^	5.42 ^a^	2.33 ^b^
LnPP_fp	81.8	76.7 ^ab^	31.6 ^a^	78.0	33.7 ^a^	5.04 ^b^	2.27 ^b^
RMSE	2.05	1.27	1.69	2.19	1.24	0.854	0.883
*p* value	NS	0.007	0.01	NS	0.0001	0.001	0.007

^ab^ means within a column under the same treatment with different superscripts are significantly different, NS= not significant, RMSE= root mean square error; IntnPP: intermediate nPP group; LnPP: low nPP group; bp: bacterial phytase; fp: fungal phytase; DM = dry matter; CP = crude protein; CF = crude fiber; EE = ether extract.

**Table 4 animals-10-00580-t004:** Carcass characteristics of 64-day-old Sasso chickens fed diets containing different non-phytate phosphorus levels with or without two types of commercial phytases.

Groups	Dressing %	Liver %	Pancreas %	Spleen %	Abdominal Fat %	Economic Efficiency %
Control	70.8	2.40 ^a^	0.200	0.165	1.93	41.2
IntnPP	69.7	2.47 ^a^	0.186	0.144	2.52	38.7
IntnPP_bp	69.9	2.43 ^a^	0.240	0.183	2.36	45.5
IntnPP_fp	70.6	2.20 ^b^	0.193	0.181	2.41	47.8
LnPP	70.4	2.22 ^b^	0.217	0.162	1.66	42.4
LnPP_bp	71.4	2.30 ^ab^	0.199	0.132	1.54	47.8
LnPP_fp	70.5	2.14 ^b^	0.222	0.154	3.00	46.1
RMSE	14.7	0.43	0.045	0.056	1.17	0.553
*p* value	NS	0.01	NS	NS	NS	NS

^ab^ means within a column under the same treatment with different superscripts are significantly different, NS= not significant, RMSE= root mean square error; IntnPP: intermediate nPP group; LnPP: low nPP group; bp: bacterial phytase; fp: fungal phytase.

**Table 5 animals-10-00580-t005:** Tibia characteristics of 64-day-old Sasso chickens fed diets containing different non-phytate phosphorus levels with or without two types of commercial phytases.

Groups	Tibia Length (mm)	Tibia Weight (g)	Tibia Diameter (mm)	Tibia Ash (%)	Tibia Calcium (%)	Tibia Phosphorus (%)
Control	11.0	7.59 ^a^	3.59 ^a^	45.1 ^a^	37.3 ^a^	14.40 ^a^
IntnPP	10.8	7.21 ^ab^	3.40 ^a^	44.8 ^a^	34.7 ^b^	13.59 ^ab^
IntnPP_bp	10.7	7.22 ^ab^	3.50 ^a^	45.1 ^a^	37.1 ^a^	15.42 ^a^
IntnPP_fp	10.6	7.03 ^b^	3.12 ^b^	45.0 ^a^	37.4 ^a^	14.43 ^ab^
LnPP	10.6	6.20 ^c^	3.18 ^b^	43.8 ^b^	35.3 ^b^	13.15 ^b^
LnPP_bp	10.9	7.04 ^b^	3.51 ^a^	44.8 ^a^	36.5 ^ab^	14.42 ^a^
LnPP_fp	11.0	7.58 ^a^	3.52 ^a^	45.0 ^a^	36.6 ^ab^	14.36 ^ab^
RMSE	0.739	1.57	0.407	1.01	0.888	1.08
*p* value	NS	0.04	0.04	0.03	0.01	0.02

^abc^ means within a row under the column treatment with different superscripts are significantly different, NS= not significant, RMSE= root mean square error, IntnPP: intermediate nPP group; LnPP: low nPP group; bp: bacterial phytase; fp: fungal phytase.

**Table 6 animals-10-00580-t006:** Chemical composition and physical characteristics of meat of 64-day-old Sasso chickens fed diets containing different non-phytate phosphorus levels with or without two types of commercial phytases.

Groups	Dry Matter %	Protein %	Lipid %	Ash %	pH	Color Intensity	Tenderness cm^2^/0.3 g	WHC cm^2^/0.3 g
Control	27.0	75.1	18.9	4.90	6.71	0.270	2.98	5.82
IntnPP	27.3	74.4	19.5	4.80	6.74	0.288	2.91	5.74
IntnPP_bp	26.4	74.4	19.6	4.60	6.75	0.295	2.89	5.70
IntnPP_fp	26.8	75.5	18.6	4.60	6.70	0.257	2.93	5.94
LnPP	26.6	75.2	18.3	4.80	6.69	0.270	2.90	5.61
LnPP_bp	26.5	74.7	18.6	5.30	6.70	0.288	2.98	5.74
LnPP_fp	26.2	74.9	18.9	4.80	6.63	0.275	2.91	5.65
RMSE	0.916	1.14	6.02	0.537	0.132	0.034	0.148	0.323
*p* value	NS	NS	NS	NS	NS	NS	NS	NS

NS= not significant, RMSE= root mean square error, pH=hydrogen power, WHC= water holding capacity; IntnPP: intermediate nPP group; LnPP: low nPP group; bp: bacterial phytase; fp: fungal phytase.

**Table 7 animals-10-00580-t007:** Metabolic profiles of 64-day-old Sasso chickens fed diets containing different non-phytate phosphorus levels with or without two types of commercial phytases.

Groups	TP, g/dL	Alb, g/dL	Glo, g/dL	TL, mg/dL	Chol, mg/dL	Calcium mg/dL	Phosphorus mg/dL	ALP U/L	AST U/L	ALT U/L
Control	4.15 ^a^	1.11 ^b^	3.00 ^a^	631	111.8 ^a^	10.9	6.70 ^a^	52.6	10.5	5.24
IntnPP	4.05 ^a^	1.55 ^a^	1.88 ^c^	692	83.7 ^b^	9.8 ^b^	6.04 ^b^	49.2	10.5	5.33
IntnPP_bp	4.36 ^a^	1.43 ^b^	3.93 ^a^	694	97.8 ^b^	12.3 ^a^	6.68 ^a^	54.1	10.4	5.57
IntnPP_fp	4.29 ^a^	1.57 ^a^	2.73 ^b^	634	105.0 ^ab^	12.4 ^a^	6.63 ^a^	50.5	10.7	5.37
LnPP	3.31 ^b^	1.67 ^a^	2.38 ^b^	691	117.2 ^a^	10.0 ^b^	6.10 ^b^	50.9	10.7	5.34
LnPP_bp	4.50 ^a^	1.69 ^a^	2.84 ^b^	689	87.2 ^b^	12.2 ^a^	6.30 ^ab^	51.5	11.0	5.57
LnPP_fp	3.86 ^ab^	1.48 ^b^	2.36 ^b^	687	119.1 ^a^	12.0 ^a^	6.64 ^a^	50.0	10.3	5.04
RMSE	0.771	0.133	0.775	53.2	17.84	4.76	0.167	3.74	3.99	0.675
*p* value	0.04	0.01	0.04	NS	0.01	0.03	0.01	NS	NS	NS

^ab^ means within a column under the same treatment with different superscripts are significantly different, NS= not significant, RMSE= root mean square error, TP: total protein, Alb: albumin; Glob: globulin; TL: total lipids; Chol: cholesterol; ALP=alkaline phosphatase; AST= asparate amino transferase; ALT= alainine amino transferase; IntnPP: intermediate nPP group; LnPP: low nPP group; bp: bacterial phytase; fp: fungal phytase.

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
