# Peer review of "Microbial and Fungal Phytases Can Affect Growth Performance, Nutrient Digestibility and Blood Profile of Broilers Fed Different Levels of Non-Phytic Phosphorous"

_animals, 2020, doi:10.3390/ani10040580_

Round 1
Reviewer 1 Report
The topic of the research is current and relevant for poultry production due to both economics and potential environmental pollution. However, in my opinion, the presentation of the obtained results requires correction.
- The introduction refers to attempts to reduce the amount of phosphorus from poultry industry waste. What were the amounts of P in the excreta of the tested chickens?
- Numerous errors in the description of the results.
- Use the same abbreviations everywhere.
- A lot of analyzes and results - no discussion, no attitude towards some in the discussion.
Other comments were placed in the text.

Author Response
Reviewer 1
The topic of the research is current and relevant for poultry production due to both economics and potential environmental pollution. However, in my opinion, the presentation of the obtained results requires correction.
A.: thank you very much for your comment
- The introduction refers to attempts to reduce the amount of phosphorus from poultry industry waste. What were the amounts of P in the excreta of the tested chickens?
A.: we did not considered here; it is well know from the literature that phytase can decrease phosphorus excursion by 18-60% depends on phytase and phosphorus level and phytic acid content of the diet (Walters, H.G., Coelho M., Coufal C.D., Lee J.T. (2019) Effects of Increasing Phytase Inclusion Levels on Broiler Performance, Nutrient Digestibility, and Bone Mineralization in Low-Phosphorus Diets. Journal of Applied Poultry Research 2019, 28:1210-1225)
- Numerous errors in the description of the results.
A.: the errors have been corrected.
- Use the same abbreviations everywhere.
A.: corrected
- A lot of analyzes and results - no discussion, no attitude towards some in the discussion.
- thank you for your comment. The discussion has been improved in several points.
-Other comments were placed in the text.
A.: all the required changes have been made
Reviewer 2 Report
Dear authors,
you provide interesting study.
I have some comments:
Abstract
L 38: ...CP and ME... please explain abbreviations when it first appear in the text.
MaM
L98: why you used unsexed broilers? Have you considered gender influence when you evaluating results?
L112: why you used NRC norm from year 1994? This is not actual nutrient requirements. This is 26 years old! Why didn't you use the nutrient requirement standard for the Sasso hybrid combination?
Table 1:
what about sand? Why you used sand in the feed mixtures? Have you analyzed chemical composition of this sand? Sand may contain Ca, P etc... Please specify sand in the text.
L124: Really was used 23:1 light-dark cycle for whole experimental period?
Are you followed the technological instructions for used hybrid? You may put information about it into the text.
L137 and L138: FCR, CPCR, MECR - explain when it first appear in the text; CPCR, g protein / g gain - g protein of feed mixutre or what? I dont understand.
Kcal is not SI units! Please used kJ.
L146 and L167: in my opinion, it is not quite right to dry the excreta (or meat) in the air-oven. Nitrogen is volatile and runs away. I think the results will not be quite accurate. We use lyophilization for excreta; and we analysed CP in whole mixed (fresh) muscle.
L147 and L166: 70 °C is right. Please correct it.
Did you use the Dumas or Kjeldahl method to determine crude protein?
Results
Table 2:
ME intake kcal - please use kJ
PCR and CCR - unify these abbreviations throughout the text. You used CPCR and MECR in L137.
Table 3:
explain abbreviations (CP, CF, EE) under table. Tables should be self-explanatory.
Table 6:
Is this Crude protein or proteins?
Table 7:
Please used ALP instead AP.
Discussion
L311, L317, L327: nnP instead nPP!
L391-393: probably delete this original journal text unless you have a own acknowledgement
Author Response
Reviewer 2
Dear authors,
you provide interesting study.
A.: thank you very much for your comment
I have some comments:
Abstract
L 38: ...CP and ME... please explain abbreviations when it first appear in the text. done
M&M
L98: why you used unsexed broilers? Have you considered gender influence when you evaluating results? Because we used adequate number of replications (5) and birds (12) within each, thus we have 60 unsexed birds within each treatment).
A: We did not consider the gender effect due to high number of animals.
L112: why you used NRC norm from year 1994? This is not actual nutrient requirements. This is 26 years old! Why didn't you use the nutrient requirement standard for the Sasso hybrid combination?
A: We agree about your point and we shall consider this in further experiment. However, as Sasso, slow grower birds, the NRC 1994, for broilers may meet their requirements.
Table 1:
what about sand? Why you used sand in the feed mixtures? Have you analyzed chemical composition of this sand? Sand may contain Ca, P etc... Please specify sand in the text.
A.: Done; sand was used at small amount (0.14-0.50%) to keep nutrients profiles of different diets similar as far as possible as, otherwise the change in feedstuffs contents may affect nutrients profiles. (lines 112-114)
L124: Really was used 23:1 light-dark cycle for whole experimental period?
A.: As indicated in the text, birds were illuminated with 24 h light cycle during the first 3 days of age and then with 23:1 light-dark cycle. A reference for this has been provided (line 124)
Are you followed the technological instructions for used hybrid? You may put information about it into the text.
A.: Broilers were breed according to https://www.hendrix-genetics.com/en/news/new-brand-promise-sasso. It has been now reported in the text (lines 127-128).
L137 and L138: FCR, CPCR, MECR - explain when it first appear in the text
A.: done
CPCR, g protein / g gain - g protein of feed mixture or what?
A.: Yes, protein and ME of feed mixture.
Kcal is not SI units! Please used kJ. corrected
L146 and L167: in my opinion, it is not quite right to dry the excreta (or meat) in the air-oven. Nitrogen is volatile and runs away. I think the results will not be quite accurate. We use lyophilization for excreta;
A.: I completely agree with you, and you like too, but we don’t have freeze drying apparatus, thus we sprayed with 4% boric acid to capture the ammonia in the form of ammonium borate, as indicated in the citied reference. This increase the accuracy of the estimation
and we analysed CP in whole mixed (fresh) muscle.
A.: We did it in equal mixture of breast and thigh muscle. In the meat (as now reported in the text) protein content was calculated by difference and not measures as CP.
L147 and L166: 70 °C is right. Please correct it. corrected
Did you use the Dumas or Kjeldahl method to determine crude protein?
A.: Kjeldahl method.
Results
Table 2:
ME intake kcal - please use kJ done
PCR and CCR - unify these abbreviations throughout the text. You used CPCR and MECR in L137.corrected
Table 3:
explain abbreviations (CP, CF, EE) under table. Tables should be self-explanatory. done
Table 6:
Is this Crude protein or proteins as protein ? protein in the meat has been calculated by difference (as now specified at line 162) and thus is protein and not crude protein
Table 7:
Please used ALP instead AP. corrected
Discussion
L311, L317, L327: nnP instead nPP! corrected
L391-393: probably delete this original journal text unless you have a own acknowledgement
A.: thank you for your suggestion
Reviewer 3 Report
Review of the manuscript no. 745866 titled "Microbial and fungal phytases can affect growth performance, nutrient digestibility and blood profile of broilers fed different levels of non-phytic phosphorous" by Youssef Attia, Fulvia Bovera, Francesco Iannaccone, Mahammed Al-Harthi, Abdulaziz A. Alaqil, Hassan Zeweil, Ali Erfat
The legitimacy of research. The benefits of using enzymes, in addition to increasing production effects, also improve their health and reduce nutrient excretion into the environment. If phytase is used, access to phytin phosphorus is obtained, which in turn reduces the amount of phosphates used in the mixtures and reduces the amount of phosphorus excreted in the faeces, and consequently prevents contamination by this element of the environment. In this aspect, undertaking this research topic is current and necessary.
Please use the following comments to improve the manuscript.
Materials and Methods
Are gender included in the study? What was the number of males and females of each replicate? What was the statistical unit, pen or individual animal?
In table 1, in the second row, for starter mixtures, the levels are: 0.50 nPP, 0.45 nPP, 0.35 nPP and should be: 0.50 nPP, 0.40 nPP, 0.30 nPP.
In the case of grower and finisher mixtures, after rounding, the P levels are practically the same, i.e. 0.40 nPP, 0.30 nPP, 0.20 nPP.
Table 1 indicates that available phosphorous was analysed, but the procedure for that assay was not described.
Why the level of P and Ca in excreta was not determined?
Statistical Analysis
Because the theory predicts the occurrence of precisely defined differences, it also makes no sense to compare each average with each other. Authors should consider whether in the case of such a planned study a more appropriate statistical analysis would be the analysis of selected planned contrasts (in other words – a priori contrasts).
Discussion
The results of significantly lower protein digestibility in the IntnPP, LownPP and LownPP_bp groups should be discussed in greater detail.
If phytates lower plasma cholesterol, why did the highest levels occur in the Control and LownPP groups?
Author Response
Reviewer 3
The legitimacy of research. The benefits of using enzymes, in addition to increasing production effects, also improve their health and reduce nutrient excretion into the environment. If phytase is used, access to phytin phosphorus is obtained, which in turn reduces the amount of phosphates used in the mixtures and reduces the amount of phosphorus excreted in the faeces, and consequently prevents contamination by this element of the environment. In this aspect, undertaking this research topic is current and necessary.
Thank you very much for your comments
Please use the following comments to improve the manuscript.
Materials and Methods
Are gender included in the study?
A.: the animals were unsexed due their high number
What was the number of males and females of each replicate?
A.: we did not know the sex ratio of each replicate
What was the statistical unit, pen or individual animal?
A.: Pen was the experimental unit, and indicated in the SA section.
In table 1, in the second row, for starter mixtures, the levels are: 0.50 nPP, 0.45 nPP, 0.35 nPP and should be: 0.50 nPP, 0.40 nPP, 0.30 nPP. corrected
In the case of grower and finisher mixtures, after rounding, the P levels are practically the same, i.e. 0.40 nPP, 0.30 nPP, 0.20 nPP.
A.: Phosphorus requirements and cost are very sensitive in broiler nutrition and can’t be round due to its effect on broiler nutrition.
Table 1 indicates that available phosphorous was analysed, but the procedure for that assay was not described.
A.: Added.
Why the level of P and Ca in excreta was not determined?
A.: We think that determination in bone was more valuable.
Statistical Analysis
Because the theory predicts the occurrence of precisely defined differences, it also makes no sense to compare each average with each other. Authors should consider whether in the case of such a planned study a more appropriate statistical analysis would be the analysis of selected planned contrasts (in other words – a priori contrasts).
A.: Dear reviewer, thank you for your comment. The contrast analysis has been performed in a first draft of the manuscript, but due to the high number of groups, it was very hard to represent the tables in an easily readable way and, in addition, the results are sometimes difficult to understand. Finally, the results of contrast analysis did not add much more to ANOVA. For all these reasons we choose the use of a simpler ANOVA.
Discussion
The results of significantly lower protein digestibility in the IntnPP, LownPP and LownPP_bp groups should be discussed in greater detail.
A.: Done. Thank you for this precious suggestion
If phytates lower plasma cholesterol, why did the highest levels occur in the Control and LownPP groups?
A.: we added some comments.
Round 2
Reviewer 3 Report
i recommend to accept this paper